# Nanominerals: Fabrication Methods, Benefits and Hazards, and Their Applications in Ruminants with Special Reference to Selenium and Zinc Nanoparticles

**DOI:** 10.3390/ani11071916

**Published:** 2021-06-28

**Authors:** Sameh A. Abdelnour, Mahmoud Alagawany, Nesrein M. Hashem, Mayada R. Farag, Etab S. Alghamdi, Faiz Ul Hassan, Rana M. Bilal, Shaaban S. Elnesr, Mahmoud A. O. Dawood, Sameer A. Nagadi, Hamada A. M. Elwan, Abeer G. ALmasoudi, Youssef A. Attia

**Affiliations:** 1Animal Production Department, Faculty of Agriculture, Zagazig University, Zagazig 44511, Egypt; saelnour@zu.edu.eg; 2Department of Poultry, Faculty of Agriculture, Zagazig University, Zagazig 44511, Egypt; 3Department of Animal and Fish Production, Faculty of Agriculture (El-Shatby), Alexandria University, Alexandria 21545, Egypt; 4Forensic Medicine and Toxicology Department, Faculty of Veterinary Medicine, Zagazig University, Zagazig 44511, Egypt; dr.mayadarf@gmail.com; 5Department of Food and Nutrition, Faculty of Human Sciences and Design, King Abdulaziz University, Jeddah 21589, Saudi Arabia; asalghamdy2@kau.edu.sa; 6Institute of Animal & Dairy Sciences, Faculty of Animal Husbandry, University of Agriculture, Faisalabad 38040, Pakistan; faizabg@gmail.com; 7University College of Veterinary and Animal Sciences, The Islamia University of Bahawalpur, Bahawalpur 63100, Pakistan; bilaladam@gmail.com; 8Poultry Production Department, Faculty of Agriculture, Fayoum University, Fayoum 63514, Egypt; ssn00@fayoum.edu.eg; 9Department of Animal Production, Faculty of Agriculture, Kafrelsheikh University, Kafrelsheikh 33516, Egypt; mahmoud.dawood@agr.kfs.edu.eg; 10Department of Agriculture, Faculty of Environmental Sciences, King Abdulaziz University, Jeddah 21589, Saudi Arabia; snagadi@kau.edu.sa; 11Animal and Poultry Production Department, Faculty of Agriculture, Minia University, El-Minya 61519, Egypt; hamadaelwan83@mu.edu.eg; 12Food Science Department, College of Science, Branch of the College at Turbah, Taif University, Taif 21944, Saudi Arabia; a.sleman@tu.edu.sa; 13Department of Animal and Poultry Production, Faculty of Agriculture, Damanhour University, Damanhour 22516, Egypt; 14The Strategic Center to Kingdom Vision Realization, King Abdulaziz University, Jeddah 21589, Saudi Arabia

**Keywords:** nanominerals, manufacturing, bioavailability, hazards, health, livestock

## Abstract

**Simple Summary:**

Nanomaterials can contribute to the sustainability of the livestock sector through improving the quantitative and qualitative production of safe, healthy, and functional animal products. Given the diverse nanotechnology applications in the animal nutrition field, the administration of nanominerals can substantially enhance the bioavailability of respective minerals by increasing cellular uptake and avoiding mineral antagonism. Nanominerals are also helpful for improving reproductive performance and assisted reproductive technologies outcomes of animals. Despite the promising positive effects of nanominerals on animal performance (growth, feed utilization, nutrient bioavailability, antioxidant status, and immune response), there are various challenges related to nanominerals, including their metabolism and fate in the animal’s body. Thus, the economic, legal, and ethical implications of nanomaterials must also be considered by the authority.

**Abstract:**

Nanotechnology is one of the major advanced technologies applied in different fields, including agriculture, livestock, medicine, and food sectors. Nanomaterials can help maintain the sustainability of the livestock sector through improving quantitative and qualitative production of safe, healthy, and functional animal products. Given the diverse nanotechnology applications in the animal nutrition field, the use of nanomaterials opens the horizon of opportunities for enhancing feed utilization and efficiency in animal production. Nanotechnology facilitates the development of nano vehicles for nutrients (including trace minerals), allowing efficient delivery to improve digestion and absorption for better nutrient metabolism and physiology. Nanominerals are interesting alternatives for inorganic and organic minerals for animals that can substantially enhance the bioavailability and reduce pollution. Nanominerals promote antioxidant activity, and improve growth performance, reproductive performance, immune response, intestinal health, and the nutritional value of animal products. Nanominerals are also helpful for improving assisted reproductive technologies (ART) outcomes by enriching media for cryopreservation of spermatozoa, oocytes, and embryos with antioxidant nanominerals. Despite the promising positive effects of nanominerals on animal performance and health, there are various challenges related to nanominerals, including their metabolism and fate in the animal’s body. Thus, the economic, legal, and ethical implications of nanomaterials must also be considered by the authority. This review highlights the benefits of including nanominerals (particularly nano-selenium and nano-zinc) in animal diets and/or cryopreservation media, focusing on modes of action, physiological effects, and the potential toxicity of their impact on human health.

## 1. Introduction

Nanotechnology is somehow a novel scientific field, and it is widely applied in many parts of our life, such as in the therapeutic field, nutrition, disease diagnosis, chemical industries, and biological research [1,2,3,4,5]. Globally, nanotechnology is an emerging and promising technology that has a formidable prospect in terms of generally revolutionizing agriculture, especially in the livestock sector [6,7]. The definition of nanotechnology is expressed as the control and comprehension of matter structure at the nanoscale range within 1–100 nanometer in size, more than 1000 times smaller than the diameter of a human hair. The first introduction of the connotation of nanotechnology was by Nobel Laureate Richard Feynman (1959), in his talk entitled “There’s plenty of room at the bottom”. He is considered the father of nanotechnology. The process of converting molecules into a nanoscale is named nanotechnology. This conversion is implemented by various chemical, physical, biological (microorganism and plants), or mixed methods [8,9,10,11]. The main idea in these approaches (converting to nanoparticles) is to induce alterations in the parent material’s fundamental chemical and physical nature. Although chemical and physical devices are further widespread for synthesizing nanoparticles, the low-dosage applications of those materials are often toxic. Moreover, they may be found in unstable status, thus critically limiting their biomedical enforcement, mainly in clinical scopes.

Moreover, this exerts chemicals into the environment [11]. Therefore, avoiding the limitation associated with the traditional synthesis methods (chemical and physical) to establish an environmentally benign procedure to create nanoparticles. Subsequently, investigators have utilized easy techniques to synthesize nanoparticles by reducing metal ions with the assistance of biological extracts, which contain flavanones, phenolics, terpenoids, amines, and alkaloids as sources of reductants [12,13]. Usually, biological approaches are safe to use and can be successfully used without other residual effects. 

A novel approach in livestock production is applying nanominerals, especially selenium (Se) and zinc (Zn), which can serve as a platform to incorporate these elements into the body. This approach enables direct transportation of active compounds to target organs, avoiding their fast degradability and encouraging several health benefits [14,15]. So far, studies have shown that the application of nanominerals in the production, immunity, and reproduction of animals is promising [6,7], but adverse effects and toxicity are also reported [14,15]. Before it is implemented in the livestock industry, the application of nanomaterials must be evaluated. Shi et al. [16] stated that nano-Se addition Se content in blood and tissues was improved. The dietary complementation of nano-Se can be used more efficiently compared to inorganic or organic Se. Inclusion of different types of selenite (sodium selenite, selenized yeast, and elemental nano-Se) increased Se levels in whole blood, serum, and tissue. Shi et al. [17] noted that nano-Se supplementation in the basal diet had enhanced the fermentation of rumen and feed. 

Many beneficial impacts of nano-Zn have been reported, such as production-promoting, enhancing animal reproductive efficiency, and antibacterial and immunomodulatory properties. Nano-Zn oxide treatment has been identified in cows, milk value has increased in clinical mastitis, milk yield has been suppressed by nano-Zn intake in dairy animals, and subclinical mastitis has been suppressed (reduced somatically counting) [6,15,18,19,20,21,22,23,24,25,26].

Researchers try to maximize the benefit from the application of nanotechnology by inserting nanominerals in animal nutrition and using their advantages to aim towards the better performance and health of animals and benefits for humans. Accordingly, this review aims to present the current knowledge related to this technology, starting with manufacturing procedures and potential applications in the ruminants’ industry, in order to impact human nutrition and the hazards for different biological systems and environmental settings.

## 2. Preparation of Nanominerals

Nanoparticles can be mainly categorized into organic and inorganic materials, based on their chemical characteristics. In the livestock sector, the nutritional values of feed can be enhanced by using organic nanoparticle (such as proteins, fats, and sugar molecules) supplementation [1]. Nutrients, in the form of nanoparticles, can be encased as nanocapsules and transported through the gastrointestinal tract (GIT) into the bloodstream, and then into many body organs, such as the brain, liver, kidney, heart, stomach, intestine, and spleen, multiplying the bioavailability of the delivered nutrients [19,20]. These capsules are proposed to transport the nutrients without any effect on appearance or taste. Encapsulated nanomaterials are combined into fodder as liposomes, micelles, and in-feed bundle systems as recognition markers, biosensors, antimicrobials, and shelf-life extenders [21]. As for inorganic nanoparticles, minerals have been used widely as nanoparticles such as calcium, magnesium, silicon dioxide, and silver nanoparticles in water and animal-related [2,22,23,24]. There are many manufacturing methods used for nanomineral fabrication with different physicochemical properties [25,26].

The intrinsic properties of nanominerals are generally determined by their shape, size, crystalline structure, composition, and morphology [27]. The shapes of the nanoparticles are numerous, including spheres, cones, rolls, worms, rectangular discs, canes, and circular or elliptical discs. These shapes strongly influence the biological behavior of nanoparticles. All these cases come up in the first, second, or third dimensions, depending on the materials used and the preparation method. The thickness and viscosity of the material used to control the particle will either be with flat or sharp endings [19]. Many factors that affect nanoparticles’ effectiveness, such as thickness and viscosity, are the base fluid viscosity, amount of nanoparticles, shape, type, diameter of particles, type, pressure, temperature, shear rate, and pH value [28,29]. Moreover, nanoparticles may show regions with several curvatures, texture concavity, and other properties that critically impact the adhesion strength that affects the effectiveness of the delivery and the efficacy of nanoparticles [19,20,27,28,29].

The notable characteristics of nanominerals are determined mainly by their shape, size, crystalline aspects, composition, morphology, and structure. Functional activities (catalytic, chemical, or biological impact) of nanominerals are strongly affected by their particles’ sizes [27]. Nanominerals have large surface areas, allowing better interface with other organic and inorganic constituents. 

With the increasing demand for nanomaterials, there is a great need to provide these materials. So, the creation of some sensitive and practical approaches to synthesize the desired nanoparticle is required. During the initial stages of nanoparticle synthesis, the main aim was to have a preferable hegemony over particle size, purity, morphology, quality, and amount [28]. As a result, different methods have been approved to synthesize nanoparticles, such as chemical, physical, and biological processes (Figure 1). In this section, we will describe the advantages and disadvantages of these techniques.

### 2.1. Physical Methods

According to many previous investigations, there is a broad framework of physical approaches for nanoparticle preparation. For instance, these approaches are similar to evaporation–condensation, which takes place by applying a boiling tube at atmospheric pressure [9]; ablation, which takes place by laser and evaporation–condensation together [29]; chemical vapor deposition; electric arc discharge; ball milling–annealing; gas-phase synthesis methods [21]; physical vapor precipitation [30]; etc. The ball mill approach is used for grinding materials into a highly fine crumb of nanosize in livestock diets. The inclusion of high energy and ball milling (HEBM) methods is quite effective (1000 times) for synthesizing the nanominerals more than traditional ball mills [31]. HEBM is a simple, inexpensive, and efficient method for the preparation of powder in bulk amounts. This processing technique, also called mechanochemical synthesis, has already been used to prepare various materials such as amorphous metallic alloys, composites, and the modification of different classes of inorganic materials [32].

Generally, a longer milling period is required for HEBM to stimulate and complete the structural alterations. However, controlled milling temperature and atmosphere must be monitored when using the HEBM. The shortfall in the synthesis of the gas phase of nanomaterials is that it usually leads to the deposition of particles with larger sizes (from 10 to 200 nm) [21]. The advantages of physical methods for the synthesis of nanominerals include the absence of solvent contamination and the maximal recovery of nanoparticles [21].

### 2.2. Chemical Methods

The chemical method represents a direct approach for synthesizing and producing materials, including several steps in the gas or liquid phase. First, the atoms’ formation can be achieved using chemical reactions under control. Thus, newly formed atoms can then undergo elementary nucleation followed by growth processes, leading to specific nanomaterials [33]. The use of chemical methods in the synthesis of nanoparticles is characterized by the extraction of nanomineral particles from modification, solvent, mass production, processing control, and stabilization of nanominerals particles from agglomeration, as well as the possibility of achieving effective and controlled bulk production compared with physical methods [34]. 

Chemical methods produce uniform and nanosized particles, but physical processes have an ample particle size range [35]. It is the most suitable for reducing the size of molecules [36]. Metal particle sizes mainly depend on the reduced capacity of reagents, where a powerful reducing reagent enhances the rate of rapid reaction and produces smaller nanometal particles [36]. However, there may be disadvantages, such as the possibility of toxicity due to hazardous chemicals during synthesis. Hence, there have been many attempts to use eco-friendly chemicals, fungal components, and plants in the production of nanomineral particles [37]. This method is called the green chemistry method. The nontoxic and eco-friendly substances used in this method include plant extracts, amino acids, starch, and glucose. Otherwise, microwave synthesis and nonchemical methods are substitutes to toxic chemical methods for the creation of nanomineral particles in a cost-effective manner and large scale [21]. 

In chemical methods, surfactants, such as polyvinyl pyrrolidone, cyclodextrin, quaternary ammonium salts, or polyvinyl alcohol, and stabilizing agents are required to inhibit the agglomeration of the metal particles [38]. Stabilizers keep the produced nanomineral particle aggregation in check, prevent the reduction of uncontrollable particle size, control particle size, and allow solubility of particles in different solvents [39]. Solvent molecules can further stabilize nanomineral particles [40]. Ligands, such as amines, phosphine, carbon monoxide, thiol, etc., can be used as stabilizers in the production of nanomineral particles through the coordination between the metal nanomineral particles and the ligand moiety [34].

### 2.3. Biosynthesis Methods

Because traditional methods are dangerous and consume more energy, there is a tendency to use “green synthesis” of nanomineral particles due to its eco-friendly, easy, and efficient nature [41] as well as the fact that it is less toxic [42]. Biosynthetic methods have emerged, using plant extracts or biological microorganisms as a simple substitute for chemical and physical processes [41]. Bacteria, viruses, algae, and plants are now used in the production of nanoparticles due to their biological advantages (low cost, nontoxic, and energy efficient) [35,42]. Biological methods were successfully used in the synthesis of different metal molecules such as gold, silver, selenium, cadmium, barium titanate, titanium, and palladium by using various plant materials [41,43], while the biosynthesis of ZnO nanoparticles was previously prepared by using *Parthenium hysterophorous* leaves [42]. Metal nanomineral particles have been synthesized with *Aloe vera, Avena sativa,* alfalfa, *Azadirachta indica*, *Sesbania drummondii*, lemongrass, the latex of *Jatropha curcas,* and papaya fruit extract [44]. The use of plant materials in nanomineral synthesis is advantageous and easier as this process is safe and straightforward. In addition, it follows one-step synthesis procedures, allowing effortless product recovery from the final solutions. Additionally, it then takes place as a one-pot synthesis and is eco-friendly, compatible with biomedical and pharmaceutical applications, economically viable, cost-effective, less time-consuming, and nontoxic, and does not need to sustain a specific culture [21]. Although there are many advantages to biological methods, there are limitations of these methods, such as maintaining the culture media, the culture condition, the difficulty in product recovery, and the period in the creation of the nanomineral particles [36]. 

Little is known about the potential synthetic methods used to produce Se nanoparticles. Anu et al. [45] reported the potential of using *Allium sativum* extracts in the green synthesis of Se nanoparticles. These extracts make Se nanoparticles spherical and crystalline with a size from 40 to 100 nm. The conventional chemically synthesized and green-fabricated Se nanoparticles were investigated to assess their cytotoxicity against Vero cells. The values of CC50 (cytotoxicity concentration 50%) indicate biologically synthesized Se nanoparticles show biocompatible features and decrease cytotoxicity compared with chemically synthesized Se nanoparticles.

## 3. Mechanism of Actions of Nanominerals

Nanoparticles, including nanominerals, can be used as functional units. Additionally, they can act as delivery means for materials associated with their surface or encapsulate inside. An animal study stated that the nanoparticle’s action mechanisms are as follows [20]: (1) raise the available surface area to connect with biological support, (2) lengthen compound residence time in GIT, (3) efficiently deliver active components to target sites in the body, (4) minimize the effect of intestinal clearance mechanisms, (5) enable efficient uptake by cells, (6) induce cross epithelial lining fenestration, e.g., liver, and (7) permeate deeply into the tissues through fine capillaries.

Recently, nanominerals have been successfully used as feed additives to fulfill livestock and poultry from the minerals. These nanoparticles are expected to possess the features of a small dose rate, better bioavailability, and stable interaction with other compounds [1,2]. Because of their low-dose use, they can be used as promoters of growth as substitutes for antibiotics, which benefit from eliminating antibiotic residues in final products, reducing environmental pollution, and producing contamination-free animal products. Additionally, these nano-additives can be integrated with micelles or capsules of protein or any other natural feed ingredient [23].

Nanoparticles enter the GIT in many ways, including inhalation and ingestion pathways and smart or oral delivery into GIT (an oral path). The different processes (absorption, metabolism, distribution, or excretion) of nanoparticles in the body rely on their physicochemical characteristics (solubility, size, and charge). For example, less than 300 nm can travel in the bloodstream, but particles smaller than 100 nm can enter into different organs and tissues [46]. 

The physiological activity of nanoparticles in the gastrointestinal tract of animals occurs through the ability of these particles for bioavailability and absorption. Nanoparticles, including nanominerals, have a larger surface-area-to-volume ratio, which provides a greater surface area for interaction with the mucosal surface, according to Corbo et al. [47]. The nanoparticulate dosage forms have shown the following advantages for gastrointestinal nanominerals delivery, owing to their smaller size: (1) easier transport through the GI tract, (2) increase in residence time of particles in the GI tract, (3) more uniform distribution and nanominerals release, (4) improved uptake into mucosal tissues and cells, and (5) specific accumulation to the site of disease, such as inflamed tissues. When a nanomineral is inserted into a biological medium such as blood or mucus, it is instantly covered with proteins adsorbed on its surface, which give it a specific “biological identity”. This protein “crown” (corona) can condition the bio-distribution as well as the possible toxicity of the nanoparticle.

Nanoparticles are usually tinier than 100 nanometers, so they easily can pass through the stomach wall and diffuse into body cells quicker than common elements with larger particle sizes. Bunglavan et al. [22] observed that the particle size of minerals, as feed additives, in the nanoparticle form is believed to be smaller than 100 nanometers. Therefore, they can pass through the stomach wall and into body cells faster than ordinary ones with larger particle sizes. 

## 4. Applications of Nanominerals in Ruminants

Microminerals can be useful for improving health and immunity, digestive system functions, microbiota homeostasis, metabolism, and reproductive performance in ruminants [48]. Additionally, they can be used for producing functional and safe animal products, maybe through eliminating the antibiotic use and increasing concentrations of trace minerals in animal products (meat and milk) required for better human health [1,6,7,14]. The health benefits and practical application of nanominerals in ruminants are illustrated in Figure 2. These effects will be displayed in detail in the following sections. 

### 4.1. Effects on Rumen Fermentation and Growth Performance

The effects of nanoparticles on growth performance, feed digestibility, and milk yield parameters in ruminants are illustrated in Table 1. Mostly dietary Zn is included in animal diets in two primary forms, i.e., organic (Zn-amino acids) and inorganic (such as ZnO and ZnSO_4_) zinc. Recently, nano-ZnO nanoparticles (ZnNPs with size within 1–100 nm) are getting more attention for use in the mineral nutrition of livestock to address dietary requirements and to promote animal growth [30]. The inclusion of ZnNPs (100 and 200 mg/kg) has increased the volatile fatty acids, microbial crude protein, and degradation of organic matter at the 6th and 12th hours of incubation period under in vitro rumen fermentation conditions [49]. Similarly, improvement in the microbial biomass production and reduction in methane emanation were recorded with 20 mg of Zn as ZnNPs compared with other higher Zn levels (40 and 60 mg/kg dry matter, DM) during an in vitro fermentation study [50]. These positive effects are also seen in vivo with adult and/or growing animals; in ewes, the dietary supplementation of ZnNPs significantly increased the digestibility of DM, organic matter, nitrogen, and crude fiber-free extract compared with Zn larger particle and control ewes [51,52]. In growing animals, the inclusion of ZnNPs in the diets of lambs enhanced the digestibility and feeding value of the diet, as revealed by the higher feed utilization efficiency observed among the experimental dietary treatments [53]. Kojouri et al. [54] reported the beneficial impact of SeNP on the weight gain of lambs during the second and fourth weeks. Male goats that received 0.03 mg/kg of nano-Se (SeNPs) had higher final body weights and average daily gains than those that received either sodium selenite or Se-yeast [16]. A positive effect has also been found in response to dietary supplementation with 50 mg Zn/kg DM, in the form of ZnO or nano-ZnO, on dry matter digestibility in Holstein calves [26]. 

Overall, these positive effects on rumen fermentation and nutrient digestibility could be ascribed to the increased surface area to volume ratio, nanoscale size, rapid and specific movement, and catalytic effectiveness. These contribute to improving absorption bioavailability of nanominerals in the GIT [55,56]. The enhancing effects of nanominerals on growth can be related to their ability to beneficially alter the gut microarchitecture of animals [57] and improve rumen fermentation, specifically fiber digestion and redox homeostasis [58,59,60].

### 4.2. Effects on Reproductive Performance

In recent years, the nanotechnology revolution has dominated all scientific fields, including farm animals’ reproduction, and facilitating particular improvements in this domain while offering many innovative interventions. The positive effects of supplementing dietary nanomaterials on reproductive performance include improving ART outcomes and addressing technical issues regarding the application of different ART in animals. Moreover, the semen purification and preservation processes have been established utilizing various nanomaterials and methods to obtain semen doses with high sperm quality [67]. Semen’s enhancement extender with antioxidant agents, such as antioxidant minerals, has been reported to upgrade the semen quality properties of cooling or post-thawing sperm cells, mainly if they are in nano-forms. Defective sperm cause failure at fertilization under both in vitro and in vivo conditions. So, maintaining perfect conditions during the storage of semen is a prerequisite for viability maintenance. During the freezing/cooling processes, sperm is preserved in synthetic extenders, which always need adjustment to maintain adequate semen quality traits. Accordingly, nanominerals have been utilized to modify semen extender properties, aiming to achieve antioxidant and antibacterial effects. Supplementation of bull semen extender with ZnNPs during cryopreservation has been detected to diminish lipid peroxidation and enhance the mitochondrial activity and functionality of sperm plasma membrane in a dose-dependent manner, without any deleterious effect on motility parameters [68]. Enrichment of semen extenders with SeNPs at a concentration of 1.0 mg/mL improved post-thawing sperm quality in Holstein bulls and, thus, the in vivo fertility rate by reducing apoptosis, lipid peroxidation, and sperm damage occurring by cryopreservation [69]. Furthermore, the addition of ZnNPs at 0.1 mg/mL to the ram semen extender enhanced total and progressive motility and the proportion of survival spermatozoa and reduced oxidative markers [70]. Similarly, the dilution of semen with a zinc nano complex resulted in a higher activity of post-thawed sperm plasma membrane integrity with a better mitochondrial activity in a dose-dependent manner. Additionally, ZnNPs exhibited no adverse effects on semen motility and raised antioxidant status. Furthermore, nanoparticles of Zn may experience beneficial actions for enhancing bovine gametes quality without affecting pregnancy rate [48]. It is likely that selenium (Se NPs) nanoparticles were employed in various research as a ROS scavenger to protect against the oxidative damage in cells’ sperm. 

The affirmative result of ZnNPs and SeNPs administrations on semen traits could be owing to the possible role of these minerals as a co-factor in the activities of several antioxidant enzymes and the protective functions against reactive oxygen species [71,72]. Both Zn and Se have been recognized as favorable for the stability and viability of spermatozoa through avoiding protein degradation and inhabited enzymes, which leads to damaged DNA of spermatozoa [67,73]. Furthermore, they maintain the stability of lysosomes, ribosomes, DNA, and RNA, which supports the survival and normal functions of sperm cells [74]. In ART related to oocytes and embryos culture, the addition of appropriate levels of nanominerals to the culture media (in vitro maturation, in vitro fertilization, and embryo culture media) can improve the developmental competence of oocytes during in vitro maturation, as well as the fertilization rate, the cleavage rate and the quality of embryos [75]. However, compared to ART related to males, studies on the effects of nanominerals on the reproductive performance of female ruminants are too limited and require further exploring.

It is well documented that minerals with potent antioxidant capacity, such as Se and Zn, are crucial trace elements in maintaining the animal’s reproductive physiology. These microminerals play an indispensable role in spermatogenesis, sperm viability, sperm cell membrane integrity, and in maintaining the chromatin structure of sperm nuclei [73,74,75,76,77]. Abaspour et al. [69] found that oral administration of Zn oxide nanoparticles (ZnNPs) at a level of 80 ppm to rams significantly enhanced sperm motility and viability rates, semen volume, sperm concentration, and the functionality of sperm membrane by 20.96, 24.03, 33.9, 11.86, and 24.4%, respectively. Moreover, it also significantly reduced sperm morphological abnormalities by 28.3%, compared with the non-supplemented group. Similarly, ZnNPs had a beneficial effect on the qualitative properties of sperm, leading to a noteworthy enhancement in some antioxidant parameters of Moghani ram seminal plasma in the non-breeding season [70]. Studies have shown positive effects of mineral administrations during different reproductive windows in females [78,79]. Many reproductive events in females, such as pregnancy and lactation, may increase the nutritional requirements of dams and change their metabolism [78]. In this respect, SeNPs supplementations to late pregnant goats significantly increased Se content in the whole blood and serum compared to selenomethionine and sodium selenite [79]. However, selenomethionine was more efficient in transferring Se into kids through the placenta and colostrum. These results refer to the role of a chemical or physical variety of Se supplementation on different physiological and biological processes.

Overall, the available literature highlights the positive roles of nanominerals on ART outcomes and on animals’ reproductive performance when supplemented with nanominerals. However, it is essential to note that the studies on in vivo models, either in males or females, are limited in drawing a complete overview for the effects of such additivies on reproductive performance and in knowing the dynamic mode of actions. The impacts of nano-Se and -Zn on the ruminant’s reproduction are found in Table 2.

### 4.3. Effects on Antioxidant Status and Health

Nanominerals may promote antioxidant activity by inhibiting the free radical’s production because of the increased surface area, leading to a higher number of active sites for scavenging an increased number of free radicals [78]. Sheep fed a basal diet containing ZnNPs exhibited a better antioxidant level [47,79]. Supplementation of SeNPs in newborn lambs promoted superoxide dismutase (SOD) levels with a concurrent reduction in thiobarbituric acid–reactive substances (TBARS) values [51].

The inclusion of Se NPs (0.6 mg Se head/day) in Khalkhali goat diets during the late stage of pregnancy significantly increased Se level in the blood (584.15μg/L) and serum (351.62 μg/L) of goats at kidding, compared with control goats (123.74 μg/L and 66.94 μg/L, respectively [79]. Moreover, it was found that introducing SeNPs to goats during the late stage of pregnancy significantly increased iron levels in the blood and serum of kids or goats and colostrum [80]. A better iron homeostasis capability was observed after the addition of SeNPs compared with other Se sources [81]. This may be due to the distinguished physicochemical features of SeNPs such as small size, large surface area, enhanced absorption via epithelial cells, and other functional properties.

The increase in blood antioxidant minerals is usually associated with particular improvements in the antioxidant status of animals and, thus, the health status of animals. Male goats offered SeNPs increased serum Se and superoxide SOD, catalase (CAT), and glutathione peroxidase (GSH-Px) compared with provided Se yeast and sodium selenite [16]. These antioxidant enzymes play the primary role of removing oxidative stress agents, such as malondialdehyde (MDA) and nitric oxide. Parallel studies by Zhan et al. [56] and Zhang et al. [80] showed that SeNPs exhibited an excellent bioavailability due to their high catalytic efficiency, low toxicity, and absorbing solid ability. These specific properties and the different absorption patterns may emphasize how nano-Se is more bioavailable than organic or inorganic Se [16].

Nanominerals also have protective effects against some physiological disorders; SeNPs showed a defensive impact on the cardiac cells from ischemia [78]. Additionally, due to the antibacterial activity of antioxidant nanominerals, some nanominerals, such as ZnO NPs, could be helpful in the prevention and curation of some bacterial-borne diseases such as sub-clinical mastitis in cows [26]. The impacts of nano-Se and nano-Zn on serum antioxidant parameters, immune response, and serum/milk Se contents in ruminants are summarized in Table 3.

## 5. Impact on the Environment and Toxicity of Nanominerals

Anthropogenic processes manufacture incidental nanomaterials as side-products [90], while engineered nanomaterials with unique properties are intentionally developed [91]. There is a review of air, water, and soil exposure from different paths. Researchers have focused on different aspects of nanomaterials, such as their innovative applications for the removal of ions and chemical molecules and contaminants, such as adsorbents, ion exchangers, and disinfectants in water and air [92,93], as well as the evaluation of their related adverse effects on human health, ecology, and the environment [94].

A specific emphasis of the topic was on papers that can cope with their health hazards and the consequences for both indoor and outdoor applications for rules and legislation [95].

Several investigations were implemented over the last few years, recommending the potential role of nanominerals such as Se and Zn in different animal nutrition and pathways [96,97,98]. Consequently, it was observed that Zn is mainly excreted from the body (because of less availability) to the environment, causing environmental contamination [73,99]. In this light, nano-Zn, as an alternative to the traditional sources of Zn, can be a great substitute in the livestock sector [71]. Therefore, nano-Zn may be used in livestock feed at lower levels to reach better findings than other Zn resources and to indirectly avoid environmental pollution. The nanoform of supplements augments the surface area that can enhance absorption and, thereby, utilize minerals, resulting in reduced dietary supplements and, ultimately, reduced feed cost and more sustainability [67]. However, the increase in nanotechnology and extensive usage of nanoparticles in everyday human life has led to worries concerning their plausible hazard impacts on live organisms and human health. The adverse impacts of nanoparticles on many cellular and molecular modifications have been well-considered, while the potential experimental toxicity needs further investigation in the studies on lab animals. Additionally, further research is essential in the future to comprehend the influence of nanominerals and their mechanisms and sites of absorption, transcript expression analysis of distribution, and mode of action [6,15].

Studies should be carried out on cellular and molecular modifications within animals to verify the effectiveness of nanominerals compared with conventional sources of minerals [2]. Furthermore, exploration should be focused on finding the ideal levels of nano-Zn in diets that can provide a better performance and reduce the hazardous impacts on the environment.

Due to the fast development of nanotechnology and future bulk manufacture of nanomaterials, there comes the need to understand, identify, and counteract any adverse health effects of these materials that may take place during manufacture, usage, or by accident [99,100,101].

The conceivable toxicological influence in both nonruminants and ruminants, along with the toxic levels, needs to be researched before they can be used in feeds. There should be systematic and comprehensive studies in order to understand the harmful effects of various doses of nanomaterials because the toxicological investigations provide various results in animal models [99].

The prime target for the use of nanomineral in animal and poultry feed is for it to be easy to absorb with no toxicity [101]. The study of the toxicity of nanominerals helps the producers and researchers in animal production and poultry to make safer decisions about the use of nanotechnologies and also increases the community’s consciousness towards nanotechnology-based applications in livestock production systems [74,102].

Selenium (Se) has been one of the crucial and necessary nutritional trace minerals for the physiological activities in the human body due to the high recovery potential of glutathione peroxidase and selenoenzymes [103,104]. However, the widespread use of Se-NPs in nanoelectronics and medication has increased the risk of their environmental contamination, which might affect living organisms, though it is useful to understand the assessment of the toxicity of Se-NPs to the biological ecosystem [105].

For instance, the nano-Se toxicity is lower than that of selenomethionine, and its toxicity is currently the lowest of all Se supplements. The toxicity of nano-Se is three times lower than that of organic Se, and seven times lower than that of inorganic Se [106].

The available literature indicates that nanoparticles of copper (CuNP) are more toxic than copper’s iconic form [20,107,108,109]. However, the explanation of the result is difficult to be made given that they indicate that CuNP safeguards proteins and DNA more effectively against nitrates and oxidation processes than Cu salts. This may be due to the clumping of nanoparticles onto agglomerates, making them less available to the body, thus reducing the damage level generated by the body. Some nanoparticles are presently described as toxic to animal and human health or the environment.

Orally feeding lambs caused severe observed renal damage (75% of animals) and mild liver toxicity (degeneration and edema in hepatocytes) when offered ZnNPs (20 mg/kg BW for 25 days) [110].

Moreover, long-term exposure to ZnNPs (250 mg/kg BW) induced liver damage in rats, which might be due to the accumulation of zinc [111]. Nanoparticles can join inside the nucleus of cells, which is the main concern that toxic nanoparticles may be able to modulate several physiobiological events of cells that may lead to cell death [112]. Higher coated ZnNPs (100 mg/kg) could damage the intestinal tract in growing pigs [113]. Studies in mice revealed that Se-NPs cause abnormal body weight, disturbances in blood biochemistry, and degeneration in hepatic and lung cells [114,115,116]. The degree of toxicity severity of nanoparticles might be attributed to mineral (organic or metallic) sources, size, shapes, dosage, synthesis methods, age of the animal, and exposure period. Lesser NPs (3–6 nm) are more easily cleared out from the kidneys compared to bigger NPs (around 30 nm), which remain in the hepatic cell [117]. Additionally, the greater size of NPs manages to stay longer in the kidney tissues due to the slower excretion machineries of glomerular filtration. This long-term retention can lead to tissue toxicity.

It was reported that the chemically synthesized ZnO NPs could be one of the possible causes of the innate toxicity of NPs, due to the chemical reaction conditions in the conventional method [118]. Accordingly, our previous work indicated that the SeNPs mainly synthesized by the biological technique at diet levels of 25 or 50 mg/kg enhanced the heat tolerance of growing rabbits [119].

Based on the literature, the toxicity of nanominerals in ruminants has not been investigated; instead, the majority of chronic or sub-chronic toxicity reports were explored using mice as an animal model [114,115,116]. In general, the applications of ZnNPs or SeNPs in animal feeding should be bordered to the precise lowest levels to avoid their negative impacts. Likewise, for better safety of NPs synthesis, the use of a microbe mediated combination approach should be considered due to its biocompatibility and the manageable shape and size of NPs, which can be realized via the optimization procedure. The best toxicological valuations have applied a cost-effective in vitro scheme, in that only precise biological paths can be confirmed under particular situations. However, the toxicologic assessments in an in vivo scheme are complex, costly, and incorporate many challenges, especially in ruminants, which have a special characterization of a microbial ecosystem [112]. Many investigations from epidemiological and experimental trials reveal that epigenetic modifications may be employed to detect toxicity produced by NPs and, more notably, to expect their potentiality of toxicity in preclinical assessment. This proposes that epigenetic modifications can be valuable indicators of NPs toxicity and can be plausible translational biomarkers for detecting unfavorable impacts of silica NPs in animals. After exposing mice cells to NPs, the damage of global DNA methylation was detected in the cells [120]. Moreover, they exhibited a substantial reduction in global cytosine DNA methylation in white blood cells from workers subjected to silica NPs [121]. Indeed, the actual molecular machinery and targets for NPs on the cellular organelles and cell membrane structure are yet to be discovered.

From the human perspective, it is recognized that alveoli contain nanoparticle deposits, and mediated mechanisms clear them through typical macrophages. A proportion of particles can translocate. This may be related to physicochemical features; however, nothing can be said regarding whether chronic exposure leads to sufficient particle accumulation to trigger disease or not [122,123,124]. Several studies reported a strong connection between high morbidity and ultrafine particles (UFP) exposure, particularly for the elderly. Moreover, recent studies indicated that there are impaction variations during particle levels and exposures for a short time as an essential cardiac activity agent.

Furthermore, specific nanoparticles may negatively induce inflammation and oxidative stress. Other materials only show toxicity at the nanoscale, impair kidney cell growth, and negatively affect cell growth and turnover [95,96,97,98,99]. According to this previous apprehension, the Toxic Substances Control Act (TSCA) was also reputable for evaluating the risk of many types of nanomaterials and to offer authority to the Environmental Protection Agency (EPA) to regulate them. Moreover, legislation administering the employ of NPs is bordered around the world in the twenty-first century, nanotechnology is one of the fast-developing areas of exploration. Therefore, a toxicity estimation of NPs is presently a demanded investigation field, the focus being NPs interaction with biological and ecological systems.

It is urgently needed that exposure to nanomaterials be considered and assessed, given that children are significantly more prone than adults when it comes to hazardous chemicals due to their larger relative body surface area. Additionally, it is essential to note that some nanoproducts are intended for use by specific subgroups, such as children and the elderly [114].

## 6. Recent Applications of Nanominerals for Human Benefits

Nanotechnology has expanded rapidly in animal health applications. Still, it is noteworthy that it is at the cutting edge of many potential human health benefits, including the production of functional food, disease diagnosis and treatment, and drug delivery [95]. Nanotechnology is shown to have beneficial applications in the human food chain, mainly through increasing the bioavailability and providing adequate amounts of essential nutrients, minerals, and vitamins in animal products consumed by humans [54,99,100]. Furthermore, the consumers’ demand for foods and their awareness has been elevated as consumers seek safe and high-quality foods with beneficial health characteristics, high sensory quality, and prolonged shelf life [110].

Improving animal production products brings many advantages to humans. The use of nano-Zn and nano-Se has been found to enhance both the quantity and quality of animal products [1]. From the human perspective, improving animal products’ contents of microelements is of particular importance. The bioavailability of minerals that originate from these sources is higher than that of minerals arising from plant sources. Considering Zn and Se, human requirements for Zn and Se are 8–15 and 55–400 μg/day, respectively, relatively higher than those provided by any nutritional source. In beef meat, the levels of Zn and Se in fresh tissue of steers average between 2.14 mg/100 g and 0.42–1.30 μg/g, respectively. Recently, nano-Zn and Se can be used as innovative and novel vehicles to improve the mineral contents of organic meats via the fortification of beef cattle diets with Se and Zn nanoelements is a candidate strategy [125]. The work on the possibility of improving the composition and quality of milk with the use of nanominerals was hardly ever produced. However, Rajendran et al. [126] used nano-zinc in feeding dairy cows and found that the use of this nanomineral reduces the number of somatic cells in cow’s milk with subclinical mastitis. Although a significant focus has been to remove potentially harmful contaminants from milk, there has also been some interest in mixing nanoparticle supplements directly into cow’s milk for human consumption [127]. The addition of combined nanopowdered oyster shells into milk to enrich the calcium content from 100 to 120 mg/mL for growing children and postmenopausal women did not negatively alter the sensory or physicochemical qualities [128].

Many studies proved the potential of supplementing nanominerals to increase mineral contents in animal products; however, most of these studies were carried out on poultry, meat and eggs [22,107,108]. More pieces of research are needed to evaluate the potential of nanominerals to change the quality and nutritional value of red meat and milk, since such food is produced by ruminants, which have a different pattern of food digestion and metabolism. However, the impact of nanominerals on health and the environment needs further investigation [129,130,131,132].

## 7. Conclusions

The global livestock sector faces the continuous pressure of ever-increasing raw material prices, including prices of minerals, which necessitates other potential sources of minerals with better bioavailability, efficacy, and lower antagonism. Nanotechnology offers potential advantages of using nanominerals in livestock nutrition, in addition to their potential impacts on human and animal health. The use of nanominerals in livestock nutrition has shown promising results in enhancing the performance, nutrient bioavailability, and immunity status of animals, and in improving the quality and composition of animal products while reducing environment-related hazards. Nevertheless, more research efforts are still needed to validate nanomaterials’ effectiveness, efficacy, and safety, in order to avoid any adverse effects on the livestock, environment, and humans.

## Figures and Tables

**Figure 1 animals-11-01916-f001:**
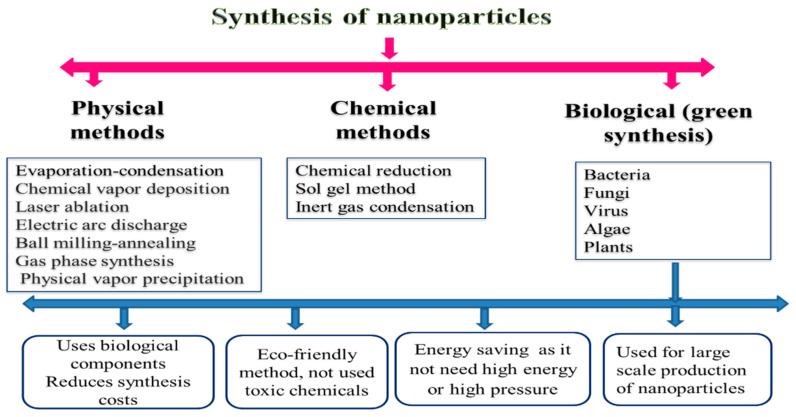
Preparation and synthesis of nanoparticles.

**Figure 2 animals-11-01916-f002:**
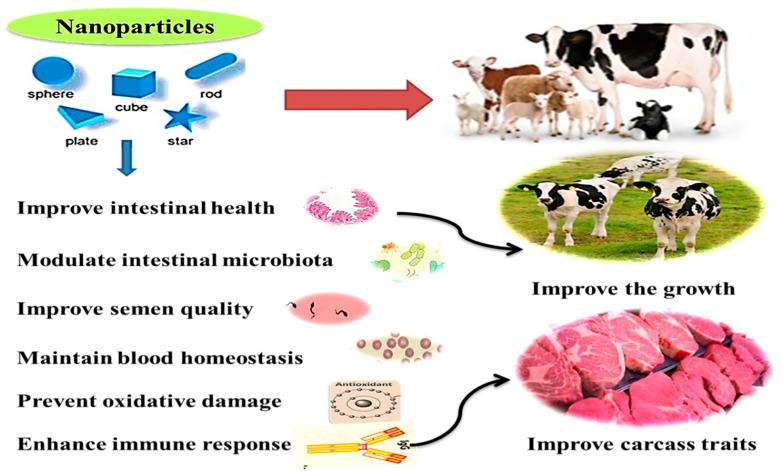
Health benefits and beneficial application of nano-minerals in ruminant.

**Table 1 animals-11-01916-t001:** Effects of nanominerals on growth performance, feed digestibility, and milk yield parameters in ruminants.

Element	Dose	Species	Major Effects
Nano-Se [17]	0, 0.3, 3 and 6 g/kg DM diet fed for 75 days	Sheep (Dorset sheep × Small Tail Han × Tan sheep)	Nano-Se at 3 g/kg DM: Increased rumen fermentation and feed digestibility.
Nano-Se and SS [61]	1 mg/kg DM diet nano-Se and SS for 10 consecutive days	Sheep (Lori–Bakhtiari breed)	Nano-Se: Exhibited better anti-oxidative effects than SS.
Nano-Se [62]	0.5 mg/kg DM diet nano-Se during gestation	Cashmere goat	Nano-Se: Improved the development of hair follicles and promoted fetal growth.
Nano-Se and SY [63]	4 mg nano-Se and YS with 4 g Se-yeast	Sheep	Nano-Se: Enhanced rumen fermentation and feed conversion efficiency as compared with YS
Nano-Se [54]	0.1 mg/kg DM diet for 60 days	Sheep (neonatal lambs)	Nano-Se: Enhanced the body growth and antioxidant parameters
Nano-Se, SS, and SY [16]	0.3 mg/kg DM diet of nano-Se, SS and SY as compared to control (0.03mg/kg Se)	Taihang black goats	ADG was higher in Nano-Se and SY than SS or control group. Nano-Se: Improved serum antioxidant enzymes (GSH-Px, SOD, and CAT) Improved serum Se contents
Nano-Se and SS [64]	0.1 mg/kg live weight of nano-Se	Sheep (Makuei breed)	Nano-Se: Enhanced weight gain Reduced the oxidative stress as compared to SS
Nano-ZnO and ZnO [52]	30 or 40 mg/kg DM diet of nano-ZnO or ZnO for pre-partum and post-partum periods	Sheep (Khorasan-Kurdish breed)	Nano-ZnO: Improved DMI, DMD, TAC in the rumen fluid Increased leukocytes and milk Zn contents.
Nano-ZnO [65]		Iranian Angora goat	Nano-ZnO: Exhibited no effect on DMI in goat kids
Nano-ZnO [49]	0, 50, 100, 200 or 400 mg/kg DM diet of nano-ZnO	In vitro ruminal fermentability	Inclusion of 100 and 200 mg of nZnO/kg: Increased the OM fermentation and VFA content Decreased the acetate-to-propionate ratio and ammonia-N
Nano-ZnO [26]	Cows exhibiting subclinical mastitis supplemented with 60 ppm inorganic zinc, zinc methionine, and nano-ZnO	Dairy cattle	Nano-ZnO: Improved milk production Reduce SCC as compared to ZnO
Nano-ZnO [66]		In vitro	Nano-ZnO: Increased the in vitro ruminal VFA contents without affecting number of protozoa.

NS = nano-Se; SS = sodium selenite; SY = selenium yeast, Nano-ZnO = nano Zinc Oxide; DM = dry matter; DMI = dry matter intake; DMD = dry matter digestibility; OM = organic matter; DWG = daily weight gain; ADG = average daily gain; FCR = feed conversion ratio TAC = total antioxidant capacity; VFA = volatile fatty acids; GSH-Px = glutathione peroxidase; SOD = superoxide dismutase, CAT = catalase.

**Table 2 animals-11-01916-t002:** Effect of nanominerals on the ruminant’s reproduction.

Element	Dose	Species	Major Effects
SeNPs [70]	1.0 µg/mL	Bull	Improved post-thawing kinematic and morphologic sperm quality Decreased apoptotic and necrotic sperm cells Improved seminal plasma & antioxidant status Increasing in vivo fertility rate
ZnNPs [68]	10^−6^, 10^−2^ molar/mL	Bull	Increasing levels of Zn-nano-complex Improved plasma membrane functionality and mitochondrial activity No deleterious effect on motility parameters
ZnNPs [48]	10^−6^, 10^−2^ M	Bull	Promoted Plasma membrane integrity Increased live spermatozoa with active mitochondria
ZnNPs [74]	50 mg/kg or 100 mg/kg	Rams	Improved epididymal semen quality, and seminal plasma
SeNPs [13]	0.3 mg/kg	Bucks	Reduced sperm abnormality rate, abnormalities in the mitochondria of the midpiece of spermatozoa Enhanced the testis Se content
ZnONPs [74]	80 ppm level	Arabic Ram	Increased the functionality of sperm membrane
CuONPs and ZnONPs Abdel-Halim et al., 2018 [75]	0, 0.4, 0.7, 1.0 or 1.5 μg/mL	In vitro maturation (IVM) of bovine oocytes	7 and 1.0 μg/mL of CuONPs or ZnONPs decreased DNA damage and increased glutathione concentrations in oocytes and cumulus, blastocyst rates 1.5 μg/mL of CuONPs or ZnO-NPs had detrimental effects on the developmental competence of bovin oocytes
SeNPs, Sodium Selenite, and L-Selenomethionine [79]	0.6 mg/head/day	Late preganat goats	Se NPs increased total Se content of the whole blood and serum L-Selenomethionine increased placental and colostral transfer of Se into kids
SeNPs [80]	1 µg/mL	Camel	Improved the progressive motility, vitality and ultrastructural morphology Decreased apoptosis of frozen semen
ZnONPs [81]	50 µg/mL	Camel	Improved sperm membrane integrity
SeNPs, [82]	2 μg/mL	Ram	Increased sperm motility
SeNPs [83]	0.5 and 1 μg/mL	Ram	Positive effects were observed on motility, acrosome protection and preservation of sperm membrane integrity
SeNPs [82,83]	0.5, 1 µg/mL	Ram	SeNPs (0.5 and 1 µg/mL) improved sperm motility, viability index, and membrane integrity

SeNPs = selenium nanoparticles, ZnNPs = zinc nanoparticles, ZnONPs = zinc oxide nanoparticles, and CuONPs = copper oxide nanoparticles.

**Table 3 animals-11-01916-t003:** Effects of nanominerals on serum antioxidant parameters, immune response and serum/milk composition in ruminants.

Element	Dose	Species	Major Effects
Nano-Se and SS [61]	Nano-Se and SS for 63 days	Sheep	Similar GSH-Px content in both sources of Se
Nano-Se, SS, and Se-Met [84]	0.6 mg/head/d for 4 weeks before parturition	Pregnant goats	Nano-Se: Increased serum Se level Se-Met: Improved Se transfer efficacy of placenta and colostrum into kid
Nano-Se [85]	0, 1 and 2 mg/kg DM diet	Sheep (male Moghani lambs)	2 mg/kg DM nano-Se: Improved the expression of liver GSH-Px and selenoprotein W1
Nano-Se and SS [86]	0.30 mg/kg of DM for one month	Dairy cows	Nano-Se: Improved milk Se and serum GSH-Px contents
Nano-Se and SS [87]	0.055 mg/kg BW for three months	Sheep (Lambs)	Nano-Se: Increased Se contents in plasma, erythrocytes, platelets, and GSH-Px activity
Nano-Se [88]	5 mg/kg BW/day	Wumeng semi-fine wool sheep	Nano-Se: Induce Se poisoning Reduced the immune and antioxidant parameters
Nano-Zn, ZnO, Zn-Met [89]	28 mg/kg DM diet	Sheep	Nano-ZnO: Increased the Zn bioavailability in rumen and blood Enhaned serum IgG Decreasing BUN contents
Nano-ZnO and ZnO [49]	Nano-ZnO supplemented at 30 or 40 mg/kg DM for pre-partum and post-partum periods	Sheep (Khorasan-Kurdish breed)	Nano-ZnO: Improved the TAC in the rumen fluid Improved milk Zn contents

NS = nano-Se; SS = sodium selenite; SY = Selenium Yeast; GSH-Px = glutathione peroxidases; DM = dry matter; IgG = immunoglobulin G; BUN = blood urea nitrogen; TAC = total antioxidant capacity.

## Data Availability

The datasets used in this study are available in text and cited in the reference section.

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
