# Peer review of "Nanominerals: Fabrication Methods, Benefits and Hazards, and Their Applications in Ruminants with Special Reference to Selenium and Zinc Nanoparticles"

_animals, 2021, doi:10.3390/ani11071916_

Round 1
Reviewer 1 Report
In general
The work titled “Nanominerals: Fabrication methods, Benefits and Hazards in Livestock Production Systems and Their Impact on Human Nutrition” concerns one of the most dynamically developing fields of science - nanobiotechnology. Unfortunately, it does not meet the criterion of a scientific publication.
The title of the work already indicates too extensive material, research in the field of the methods producing nanoparticles, benefits and hazardous in animal production and the impact on human nutrition are numerous. It should be remembered that there are many nanoparticles of mineral elements and compounds, and they include not only Se and ZnO but also Ca, P, Mg, Cu, Fe, Mn and other elements and their compounds. The authors limited themselves to discussing only Zn and Se compounds. The production methods, on the other hand, were related to the production of all nanostructures, without focusing on the most important ones from the animal production point of view. In addition, benefits were primarily discussed and the topic of harmful effects of nanoparticles and their causes was minimized.
The authors limited themselves to discussing selected issues related to ruminants, and even these passages have not been fully discussed. Livestock also includes several other important species, including pigs and poultry. These animals were not even mentioned. The topic of the impact on human nutrition has been completely marginalized and completely omitted.
The work is very chaotic, contains excerpts from the cited studies that have been discussed very superficially. The whole and individual subsections are inconsistent, they do not form a logical thread but rather a random whole. The authors did not comment on the biochemical or physiological background of the activity of nanoparticles. Most of the work is on the level of a popular publication, not a scientific one.
Abstract: Many repetitions from the Simple summary, no generalization of the whole, no information about the content contained in the publication, only about details. Detailed examples "for example" L-53, L-56 are redundant. The most important content and problems that are the content of the work were not presented.
Introduction: The text is chaotic, many sentences are not related to each other, no introduction to the subject of the publication. Two specific examples of Zn and Se nanoparticles reappear. The sentence - L-108 - 109 is not true, research and work on nanotechnology in animal production and biotechnology are numerous. Many of them have not been cited. It seems that the authors wanted to write about everything, but only wrote about Zn and Se nanoparticles and ruminants.
Chapter 2: The text is also disorderly, illogically presented, it concerns random issues, it does not organize the material that will be discussed in the following chapters. The authors, citing the literature, mix forecasts and research plans with real results and applications. Most of the text in the introductory section of this chapter is off topic. Subsections 2.1, 2.2 and 2.3 are a great simplification of the topic. The biological methods of synthesizing nanoparticles are in fact chemical methods, only on a very small scale. The same reduction process that occurs in the plant can be used in a test-tube, and the harmfulness of this reaction is only due to its efficiency (higher in vitro).
Chapter 3: The authors only referred to the positive aspects of nanoparticle activity, and there are many known negative effects of their action. The physiological activity of nanoparticles in the gastrointestinal tract of animals, including ruminant vs monogastric, has not been discussed. Anyway, this topic is extensive and could constitute a separate work in itself. The authors did not address the problem of agglomeration or the formation of the so-called "protein crown" on the surface of nanoparticles during their self-assembly with proteins (abundant in the gastrointestinal tract).
Chapter 4: A major drawback of the work is the general reference to nanoparticles. Individual nanoparticles are different element and their activity may be completely different (even opposite). The results of other authors are presented in a very simplified manner without mentioning the mechanisms (e.g. L-261, 262).
The numbering 3 x 4.2 was mixed up in the text. Nevertheless, only ZnO and Se nanoparticles are discussed in these subsections, as reflected in the tables. Tables (1, 2 and 3) and the information gathered therein are the only valuable element of the publication. In the subsection "Effects on Reproductive Performance", the authors only discussed the influence of ZnO and Se nanoparticles on sperm preservation and quality, ignoring other aspects of the impact of mineral nanoparticles' on reproduction.
Chapter 5: The topic of this chapter could be the title of the textbook. Therefore, it is not surprising that it contains little information and mainly generalities, phrases, and repeated sentences.
Chapter 6: Such an extensive topic is covered on 0.5 pages, so it is difficult to expect that it will fully reflect the relevant issues. Many literature items are not covered.
Conclusions: Too general, chaotic.
Author Response
In general
The work titled "Nanominerals: Fabrication methods, Benefits and Hazards in Livestock Production Systems and Their Impact on Human Nutrition" concern one of the most dynamically developing fields of science - nanobiotechnology. Unfortunately, it does not meet the criterion of a scientific publication.
Response: We have done our best to improve the manuscript considering your comments and valuable suggestions. The manuscript was enhanced and reconstructed, and new references were shown, and great improvements were made as far as possible according to the valuable suggestions and comments of the reviewers.
Comment: The work title already indicates too extensive material, research in the field of the methods producing nanoparticles, benefits and hazardous in animal production and the impact on human nutrition are numerous. It should be remembered that there are many nanoparticles of mineral elements and compounds, and they include not only Se and ZnO but also Ca, P, Mg, Cu, Fe, Mn, and other elements and their compounds. However, the authors limited themselves to discussing only Zn and Se compounds. On the other hand, the production methods were related to the production of all nanostructures without focusing on the most important ones from the animal production point of view. In addition, benefits were primarily discussed, and the topic of harmful effects of nanoparticles and their causes was minimized.
Response: Thank you for your comment, which supports our paper. You are right, but to be honest, in this review, we focused only on Nanomaterials: Fabrication methods, Benefits and Hazards, and Their Applications in Ruminants with particular Reference to Selenium and Zinc Nanoparticles. This, not our choice but this what is available in the literature. These two minerals are hugely used in animal feed nutrition due to their essential effects on productivity and health. Other Nano minerals have limited uses; for example, nanosilver is used as an antimicrobial agent; however, most available studies are in vitro studies and out of our review scope. Therefore, to be more explicit, we changed the title to make it more suitable for the content. Also, regarding the harmful effects of nanoparticles, we discussed this part under the 5—impact on the Environment and Toxicity of Nano-minerals.
Comment: The authors limited themselves to discussing selected issues related to ruminants, and even these passages have not been thoroughly examined. Livestock also includes several other important species, including pigs and poultry. These animals were not even mentioned. The topic of the impact on human nutrition has been completely marginalized and wholly omitted.
Response: yes, you are right. We focused only on the applications of nano minerals in ruminants. As there are recent available reviews/studies on the applications of nano-materials in poultry and monogastric animals. Thus, we changed the title to make it more suitable for the content. Also, regarding the impact on human nutrition, we did not mean the effects of nano minerals on human nutrition. Still, we meant the effects of nano minerals on animal products and their relationship to human health. Furthermore, we improved this part and added some valuable parts.
Comment: The work is very chaotic and contains excerpts from the cited studies discussed superficially. The whole and individual subsections are inconsistent, and they do not form a logical thread but rather a random whole. The authors did not comment on the biochemical or physiological background of the activity of nanoparticles. Most of the work is on the level of a popular publication, not a scientific one.
Response: Thank you for your deep speculation and full review. The authors added some details and scientific background when relevant. Please, check lines 268-280; 394-408, and 514-520.
Comment: Abstract: Many repetitions from the Simple summary, no generalization of the whole, no information about the content contained in the publication, only about details. Detailed examples "for example" L-53, L-56 are redundant. The most critical content and problems that are the content of the work were not presented.
Response: Redundant sentences have been deleted. Manuscript elements have been included in the abstract and have been improved.
Comment: Introduction: The text is chaotic; many sentences are not related to each other, no introduction to the subject of the publication. Two specific examples of Zn and Se nanoparticles reappear. The sentence - L-108 - 109 is not true. Research and work on nanotechnology in animal production and biotechnology are numerous. Many of them have not been cited. It seems that the authors wanted to write about everything but only wrote about Zn and Se nanoparticles and ruminants.
Response: The introduction has been improved, and incorrect sentences removed. We changed the title to make it more suitable for the content.
Comment: Chapter 2: The text is also disorderly, illogically presented; it concerns random issues, it does not organize the material that will be discussed in the following chapters. The authors, citing the literature, mix forecasts and research plans with real results and applications. Most of the text in the introductory section of this chapter is off-topic. Subsections 2.1, 2.2, and 2.3 are a great simplification of the topic. The biological methods of synthesizing nanoparticles are chemical methods, only on a very small scale. The same reduction process that occurs in the plant can be used in a test-tube, and the harmfulness of this reaction is only due to its efficiency (higher in vitro).
Response: Thank you for your comment. You may have some rights to classify biological methods as chemical methods because they may share the use of chemicals. However, in biological methods, many attempts are paid to substitute chemical compounds such as potent reducing agents with other natural plant extracts with the same reducing ability. This may mitigate health risks and hazards related to the extensive use of synthetic chemicals. Finally, according to our knowledge, these were the available data in method classification for nanoparticle preparation, including nano minerals.
Comment: Chapter 3: The authors only referred to the positive aspects of nanoparticle activity, and there are many known adverse effects of their action. The physiological activity of nanoparticles in the gastrointestinal tract of animals, including ruminant vs. monogastric, has not been discussed. Anyway, this topic is extensive and could constitute a separate work in itself. The authors did not address the problem of agglomeration or the formation of the so-called "protein crown" on the surface of nanoparticles during their self-assembly with proteins (abundant in the gastrointestinal tract).
Response: We thank you for your comment, which supports the outcomes of our paper. The physiological activity of nanoparticles in the gastrointestinal tract was shown in the manuscript. Also, the problem of agglomeration or the formation of the so-called "protein crown" was mentioned.
Comment: Chapter 4: A major drawback of the work is the general reference to nanoparticles. Individual nanoparticles are different elements, and their activity may be completely different (even the opposite). Furthermore, the results of other authors are presented in a very simplified manner without mentioning the mechanisms (e.g. L-261, 262).
Response: in chapter 4: we discussed the applications of nano minerals in ruminants and their effect on health and immunity, digestive system functions, microbiota homeostasis, metabolism, and reproductive performance in ruminants. Also, based on valuable comments, we improved chapter 3 (Mechanism of Action of Nano minerals) and some useful parts. Thanks for your efforts and for giving us a chance to improve our paper.
Comment: The numbering 3 x 4.2 was mixed up in the text. Nevertheless, only ZnO and Se nanoparticles are discussed in these subsections, as reflected in the tables. Tables (1, 2, and 3) and the information gathered therein are the only valuable publication element. In the subsection "Effects on Reproductive Performance," the authors only discussed the influence of ZnO and Se nanoparticles on sperm preservation and quality, ignoring other aspects of the impact of mineral nanoparticles' on reproduction.
Response: Numbering has been correctly modified. Also, based on your opinion and the second reviewer's opinion, we changed the title to make it more suitable for the content as "nano minerals: Fabrication methods, Benefits and Hazards, and Their Applications in Ruminants with Special Reference to Selenium and Zinc Nanoparticles."
Regarding the studies shown for the effects of nano minerals on reproduction, the authors did another search and explored other available studies. The available studies were examined and presented (Table 2). Please, note that the majority of published data are on semen storage and purifications. Studies on ART in females and in vivo studies are too limited.
Comment: Chapter 5: The topic of this chapter could be the title of the textbook. Therefore, it is not surprising that it contains little information and mainly generalities, phrases, and repeated sentences.
Response: several improvements and adjustments have been made. Also, different paragraphs have been added to improve this chapter.
Comment: Chapter 6: Such an extensive topic is covered on 0.5 pages, so it is difficult to expect that it will fully reflect the relevant issues. Many literature items are not covered.
Response: Some references that support this section were supplemented. If there are references that we did not use, please point them to us to improve the manuscript. Our goals to show that we focus on the opportunity to enhance minerals in milk and meat (animal products only) using nano minerals, which is pretty covered in the literature.
Comment: Conclusions: Too general, chaotic.
Response: We did our best and Improved as far as possible, and we welcome further suggestions and valuable comments, if any.
Reviewer 2 Report
This review is a hymn on he use of nano-Zink and nano-selenium in animal hubandry. The authors present ample evidence that these nano-materials are a must in animal nutrition. I guess that their enthusiasm is not shared by all readers. However, it is clear that nano-materials can be an advantage in animal production. Possible deleterious effects are given a much smaller space, and if these materials are toxic its a rare case or a different nano-material. I like it if authors do not hide their opinion behind some spurious data. This is legitimate and highly welcome. The background of the authors is applied science and their focus is the improvement (increase of quality and quantity) of animal production. No wonder that the knowledge of adverse effects is limited. Still the review is nice to read and a good basis for discussion. Last but not least the authors give an outlook of future develpments in their field. And all doubts fade away behind the fact that more research is necessary.
Author Response
Comment: This review is a hymn on he uses of nano-Zink and nano-selenium in animal husbandry. The authors present ample evidence that these nano-materials are a must in animal nutrition. I guess that their enthusiasm is not shared by all readers. However, it is clear that nano-materials can be an advantage in animal production. Possible deleterious effects are given a much smaller space, and if these materials are toxic, its a rare case or a different nano-material. I like it if authors do not hide their opinion behind spurious data. This is legitimate and highly welcome. The background of the authors is applied science, and their focus is the improvement (an increase of quality and quantity) of animal production. No wonder that the knowledge of adverse effects is limited. Still, the review is nice to read and a good basis for discussion. Last but not least, the authors give an outlook of future developments in their field. And all doubts fade away behind the fact that more research is necessary.
Response: Thank you very much for these supportive comments; we highly appreciate the tremendous effort and valuable time you spent reviewing our paper. This manuscript reviews more references on nano-zinc and nano selenium in livestock. These two nano-elements are the most used in recent studies and their essential effects on animal health, immunity, productivity, and reproduction. However, for this MS, we only refer to the toxicity of nanomaterials, it needs a separate paper to explain it in detail. Thus, we thank you for this clarification, which will open the way for us to work with a toxicologist.
Reviewer 3 Report
I read this article with interest, which deals with the very topical subject of nanomaterials. It is an interesting alternative to introducing minerals into the organisms of animals (and perhaps humans in the future).
However, several points in this article caught my attention.
After the title, which reads: "Nanominerals: Fabrication methods, Benefits and Hazards in Livestock Production Systems and Their Impact on Human Nutrition" I expected a broader development of the topic related to methods of their production and influence on the human body. My guess is that there may not be enough literature to complete these points. If so, maybe the title should be changed to match the content of the article?
The sentences in line 118-120 and 140-142 have a similar meaning (for me) and the sense of these sentences in the meaning should be more differentiated.
Line 132: The sentence "The thickness and viscosity of the material used to control the particle will either be with flat or sharp endings" requires some details to explain to the reader how the dependencies are presented.
Line 134-135: "Nanoparticles may show regions with several curvatures, texture concavity, and other properties" - please elaborate and describe this idea.
Line 143: I don't understand this part of the sentence: "With the increasing demand for nanomaterials, nanomaterial had huge indigence, ...". Please edit it.
Line 160: Is it possible to refine the "High Energy and Ball Milling (HEBM) methods"?
Part "2.2. Biosynthesis Methods" - it is advisable to supplement the description with details of the processes occurring in this type of preparation of nanominerals methods.
Line 228: "The nanoparticles action's mechanisms are as follows: ..." - please clarify which organisms these actions affect: animals or humans?
Table 1, "Major effect" column - only general reactions / results are given, can't any details (or percentages) be given?
Editorial (small) notes:
Figure 2: one of the labels reads as follows: "Change intestinal microbita". Is it the word "microbiota"?
line 501: title: "Conclusion and Recommendation". Rather, it is a summary - a conclusion. Who would be the addressee of these recommendations?
line 770: please delete 110, because in the text reference numbers end with 109.
Author Response
Comment: I read this article with interest, which deals with the very topical subject of nanomaterials. It is an interesting alternative to introducing minerals into the organisms of animals (and perhaps humans in the future).
Response: Thank you very much for your extraordinary efforts; several adjustments have been made to improve our paper; thank you again for trying to improve our paper.
Comment: However, several points in this article caught my attention.
After the title, which reads: "Nanominerals: Fabrication methods, Benefits and Hazards in Livestock Production Systems and Their Impact on Human Nutrition," I expected a broader development of the topic related to methods of their production and influence on the human body. My guess is that there may not be enough literature to complete these points. If so, maybe the title should be changed to match the content of the article?
Response: Thank you very much for your great efforts; based on your opinion, we changed the title to make it more suitable for the content as "Nanomaterials: Fabrication methods, Benefits and Hazards, and Their Applications in Ruminants with Special Reference to Selenium and Zinc Nanoparticles."
Comment: The sentences in lines 118-120 and 140-142 have a similar meaning (for me) and the sense of these sentences in the purpose should be more differentiated.
Response: The two sentences were merged to prevent repetition.
Comment: Line 132: The sentence "The thickness and viscosity of the material used to control the particle will either be with flat or sharp endings" requires some details to explain to the reader how the dependencies are presented.
Response: more details are given in the text as there many factors, which affect the thickness and viscosity are the base fluid viscosity, amount of nanoparticles, shape, type, and diameter of particles type, pressure, temperature, shear rate, and pH value. This could affect the properties and effectiveness of the nanoparticles.
Comment: Line 134-135: "Nanoparticles may show regions with several curvatures, texture concavity, and other properties" - please elaborate and describe this idea.
Response: this was elaborated in the text as Nanoparticles may show regions with several curvatures, texture concavity, and other properties which critically impact the adhesion strength that affects the effectiveness of the delivery and the efficacy of nanoparticles
Comment: Line 143: I don't understand this part of the sentence: "With the increasing demand for nanomaterials, nanomaterial had huge indigence, ...". Please edit it.
Response: Edited
Comment: Line 160: Is it possible to refine the "High Energy and Ball Milling (HEBM) methods"?
Response: Yes, done.
Comment: Part "2.2. Biosynthesis Methods" - it is advisable to supplement the description with details of the processes occurring in this type of preparation of nano minerals methods.
Response: this part has been improved.
Comment: Line 228: "The nanoparticles action's mechanisms are as follows: ..." - please clarify which organisms these actions affect: animals or humans?
Response: It was clarified that the study was conducted on animals
Comment: Table 1, "Major effect" column - only general reactions/results are given; can't any details (or percentages) be given?
Response: Thank you for your comment; we only refer to the most important results regarding the target point and avoid showing a massive table with more details disrupting the reader.
Editorial (small) notes:
Comment: Figure 2: one of the labels reads as follows: "Change intestinal microbita". Is it the word "microbiota"?
Response: Corrected, and we improved the quality of the figure.
Comment: line 501: title: "Conclusion and Recommendation". Rather, it is a summary - a conclusion. Who would be the addressee of these recommendations?
Response: Modified to Conclusion only. If the Conclusion contains advice, it is for researchers and specialists in the field
Comment: line 770: please delete 110 because, in the text, reference numbers end with 109.
Response: References was adjusted.